# Therapeutic Targeting of Fumaryl Acetoacetate Hydrolase in Hereditary Tyrosinemia Type I

**DOI:** 10.3390/ijms22041789

**Published:** 2021-02-11

**Authors:** Jon Gil-Martínez, Iratxe Macias, Luca Unione, Ganeko Bernardo-Seisdedos, Fernando Lopitz-Otsoa, David Fernandez-Ramos, Ana Lain, Arantza Sanz-Parra, José M Mato, Oscar Millet

**Affiliations:** 1Precision Medicine and Metabolism Laboratory, CIC bioGUNE, Bizkaia Technology Park, Bld. 800, Derio, 48160 Bizkaia, Spain; jgil@cicbiogune.es (J.G.-M.); iratxe.maciasgarcia@osakidetza.eus (I.M.); lunione@cicbiogune.es (L.U.); flopitz@cicbiogune.es (F.L.-O.); dfernandez.ciberehd@cicbiogune.es (D.F.-R.); alain@cicbiogune.es (A.L.); asanz@cicbiogune.es (A.S.-P.); director@cicbiogune.es (J.M.M.); 2ATLAS Molecular Pharma, Bizkaia Technology Park, Bld. 800, 48160 Derio, 48160 Bizkaia, Spain; gbernardo.atlas@cicbiogune.es

**Keywords:** tyrosinemia, tyrosinemia type I, fumaryl acetoacetate hydrolase, rare disease, metabolic rare disease, drug discovery, tyrosine, nuclear magnetic resonance

## Abstract

Fumarylacetoacetate hydrolase (FAH) is the fifth enzyme in the tyrosine catabolism pathway. A deficiency in human FAH leads to hereditary tyrosinemia type I (HT1), an autosomal recessive disorder that results in the accumulation of toxic metabolites such as succinylacetone, maleylacetoacetate, and fumarylacetoacetate in the liver and kidney, among other tissues. The disease is severe and, when untreated, it can lead to death. A low tyrosine diet combined with the herbicidal nitisinone constitutes the only available therapy, but this treatment is not devoid of secondary effects and long-term complications. In this study, we targeted FAH for the first-time to discover new chemical modulators that act as pharmacological chaperones, directly associating with this enzyme. After screening several thousand compounds and subsequent chemical redesign, we found a set of reversible inhibitors that associate with FAH close to the active site and stabilize the (active) dimeric species, as demonstrated by NMR spectroscopy. Importantly, the inhibitors are also able to partially restore the normal phenotype in a newly developed cellular model of HT1.

## 1. Introduction

Tyrosinemia type I (HT1) is a rare autosomal recessive genetic disease caused by a deficiency in fumarylacetoacetate hydrolase (FAH, EC 3.7.1.2), the last enzyme in the tyrosine catabolism pathway [1]. Failure to properly break down tyrosine leads to the abnormal accumulation of toxic metabolites, resulting in severe liver and kidney dysfunction. The pathology has an average estimated incidence of 1 in 100,000 births worldwide, but the highest incidence is recorded in specific geographical areas such as Quebec [2]. Symptomatology appears in the first few months of life and includes failure to thrive, hepatomegaly, and jaundice, which can lead to hepatocellular carcinoma and/or renal tubular dysfunction [3]. Untreated children often do not survive past the age of ten.

The current treatment of HT1 combines a low protein diet that contains limited amounts of phenylalanine and tyrosine combined with the herbicide nitisinone (2-(2-nitro-4-trifluoro-methylbenzoil)-1,3-cyclohexanedione), a reversible inhibitor of 4-hydroxyphenylpyruvate dioxygenase, the second enzyme of the tyrosine catabolism [4]. This treatment is effective but it does not supply for tyrosine degradation, and the most common adverse reactions for nitisinone are associated with the elevated tyrosine levels: a bloated abdomen, dark urine, abdominal pain and weight loss [5]. Moreover, progression to cirrhosis, liver failure, and potential hepatocellular carcinoma is still possible, and liver transplantation is needed in patients in whom nitisinone fails [6]. Altogether, there is still demand for alternative pharmacological treatments to replace and/or complement the existing ones.

At the molecular level, FAH is a cytosolic homodimer with two 46 kDa subunits formed by a 120-residue N-terminal domain (N-term) and a 300-residue C-terminal domain (C-term). N-term has an SH3-like fold and is supposed to play a regulatory role [7,8], while C-term is defined by a new β-sandwich roll structure that is implicated in metal-ion binding and catalysis, participating in intermolecular interactions at the dimer interface [9]. To date, around 100 mutations have been reported to cause HT1: 45 missense mutations, 23 splice defects, 13 nonsense mutations, 10 deletions and 4 frameshift alterations [10]. C-term contains the majority of the pathogenic defects (69 vs. 9). Recently, we showed that the dimer of FAH is the functionally pertinent species while monomeric FAH tends to aggregate at physiological conditions [11]. Impaired protein homeostasis is exacerbated in many of the missense mutations, providing a molecular mechanism for HT1 disease.

In this context, an attractive alternative way to regulate tyrosine catabolism is by means of pharmacological chaperones, which are chemical substrates or modulators that usually bind to the partially folded intermediate, stabilize the protein, and allow it to complete the folding process to yield a functional protein [12]. These chemical entities have successfully reduced clinical symptoms of disease by slowing down or inhibiting the tendency of different proteins to aggregate, resulting in detectable enzyme levels in the cell (see Table 1 in the review by Loo [13]). Pharmacological chaperones usually target the binding site of the enzyme acting as reversible inhibitors, but this is not always the case and we recently demonstrated that the off-patent synthetic antimicrobial ciclopirox acts as an allosteric pharmacological chaperone, targeting variants of the enzyme uroporphyrinogen III synthase with impaired homeostasis [14].

In this study, we investigated the putative therapeutic potential of the chemical modulation of the tyrosine catabolism, targeting FAH. To that end, we screened several thousand chemical entities for the in vitro stabilization and intracellular optimization of the homeostatic properties of FAH, followed by a thorough biophysical and biochemical analysis of the selected hits. As a result, we have found a set of reversible inhibitors that associate to FAH with moderate affinity at the active site. NMR experiments demonstrate that the inhibitors stabilize the dimeric (active) species, and we exploited the simultaneous binding of some fragments to a second site to increase the chemical potency in a second generation of compounds, with low micromolar affinity for FAH. These molecules do not just act as in vitro inhibitors, but they are also capable of partially restoring the non-pathogenic phenotype in a newly developed cellular model of HT1.

## 2. Results

### 2.1. The Dimerization Interface Is Druggable

Computational docking was used to investigate the capability of FAH to associate molecules and to detect the putative protein binding sites for the ligands. First, a molecular dynamics simulation (MD) was run for 100 ns on the high-resolution dimeric FAH structure (PDB code 1QCN [15]) from Mus musculus (with 89% identity to *Homo sapiens*). Four conformations were selected along the MD trajectory to adequately represent the protein flexibility and dynamics. Docking experiments were performed to test a library of 20,000 compounds from three main virtual libraries on the four selected conformations of FAH. Despite the large chemical diversity of the screened compounds, computational docking enabled identifying up to four independent druggable sites of FAH. The vast majority of the molecules from the library accommodate either in the pockets of the active site (86%) or in the dimerization interface (29% but with only a 2% of non-overlapping binders at this site) (Appendix A). A minor subset of molecules binds to other regions of the protein, such as the outer region of the dimerization interface (8%) or the interface between the N- and the C-domain of the protein (4%). These results suggest that the dimerization interface is druggable and can be targeted by organic molecules, paving the way for the molecular modulation of the dimerization (and aggregation) processes.

### 2.2. Small Molecules to Stabilize the Dimer Conformation of FAH

A protein thermal shift assay (PTS), performed in a semi high-throughput mode, was used to screen a library of 2500 chemical fragments to identify molecular entities capable to enhance FAH thermostability (changes in the protein melting temperature, T_m_). As previously described [14,16], in PTS, thermal stability is monitored by means of a dye (Protein Thermal Shift^TM^ dye). Under native conditions, this dye is naturally quenched, while, upon temperature denaturation, it reacts with the hydrophobic core of the protein that becomes exposed, resulting in a significant increase in fluorescence emission. The target protein selected for the screening was D233V-FAH, which shows decreased stability (with a ΔT_m_ of −23 °C as compared to WT FAH) and enhanced tendency towards aggregation, but preserves the stability of the dimeric species [11]. These features indicate that D233V-FAH is a valid target to identify molecules that stabilize the protein when binding at the dimeric interface. For simplicity, the screening was performed at a protein concentration (1 µM) where the monomer is the dominant species, so a one-sigmoid thermogram was found (Figure 1A). For each compound, the test was repeated four times to ensure statistical significance (Figure 1B). The screening resulted in a total of 52 compounds (2.08% of the library) that significantly increased the T_m_ of D233V-FAH between 2.0 and 4.5 °C. The stability increases are moderate, consistent with the chemical nature of the ligands (fragments with MW < 300 Da), which limits the maximum number of stabilizing intermolecular contacts.

### 2.3. Functional Screening to Improve Intracellular FAH Homeostasis

The stability assay was complemented with a functional assay. As shown previously [11], a plasmid stably expressing GFP-tagged FAH is suitable to monitor intracellular FAH homeostasis. In this context, a second in cellulo screening used human M1 cells stably expressing V166G-FAH-GFP, which were tested over the same library of 2500 fragment compounds, monitoring the GFP fluorescence by high-contrast automated fluorescent microscope [14,17,18]. V166G-FAH is particularly suitable for this experiment, since it shows no basal fluorescence, while, upon treatment with the proteasomal inhibitor MG-132, 19% of the cells become fluorescent, as determined by flow cytometry (Figure 2A). In the presence of an inactive compound, no fluorescence was observed (Figure 2B). Yet, when analyzing the entire set, 113 compounds (4.52% of the library) were found to significantly increase GFP fluorescence. Figure 2C shows a representative example of compound-induced FAH activation, with more than 80% of fluorescent cells.

An increase in the fluorescence indicates a rise in the FAH intracellular concentration, and it is tentatively interpreted as a compound-induced increase in the intracellular homeostasis. Yet, many other causes may also produce it and we crossed the results obtained from the independent stability and functional assays for validation. Interestingly, 90% of the compounds that gave positive results in both assays were also docked at the active site of the enzyme. A first generation of three selected compounds (G1, Figure 2) showed a statistically significant in vitro stabilization of D233V-FAH (Figure 2E) with an increase in the intracellular accumulation of V166G-FAH-GFP in M1 cells (Figure 2D). These compounds are considered as putative pharmacological chaperones acting on FAH, and were further evaluated.

To investigate how general the compound-induced FAH homeostasis modulation is, we prepared three new stably expressing M1 clones (A35T-, W234G-, and T294P-FAH-GFP) that were tested over the three selected compounds (Figure 2D). Considering the inherent intrinsic variability in FAH-GFP expression between experiments, the results from the different cell lines are reasonably consistent and the active compounds elevate the intracellular FAH levels for all the clones tested (Figure 2D).

### 2.4. Organic Molecules Modulate FAH Catalytic Activity In Vitro

Next, we checked whether the set of compounds that stabilize FAH had any effect on the catalytic activity of the enzyme. If the compounds acted as pharmacological chaperones (Appendix A), they should prevent protein destabilization, increasing the lifetime of the active protein and ultimately the enzyme activity. V166G-FAH was selected since it retains 68% of WT FAH catalytic activity. To measure FAH activity, we quantified the residual amount of fumaryl acetoacetate (FAA) by HPLC after a given time for the enzymatic reaction and compared it to the control experiment. In this experiment, an increase/decrease in the FAA signal upon compound addition implies an inhibition/activation of the FAH enzyme (Figure 2F), and this was quantified, relative to the control reaction, for the three compounds tested at two different concentrations. At a ratio of 1:15 (FAH:compound) most of the compounds activate the enzyme (Figure 2E), a result consistent with their role as a pharmacological chaperone. On the other hand, at a ratio of 1:30 (FAH:compound), all the compounds but one inhibit the enzymatic reaction, suggesting that the compounds are indeed reversible inhibitors that associate to the enzyme at positions that compete with the substrate: at low concentrations, the stabilization effect dominates, while, at high concentrations, enzyme inhibition takes place as the predominant mechanism.

The compound G1.30.B10 (Figure 3A) enhances the activity of V166G-FAH at all ratios, indicating less competition with the substrate for binding, and we investigated it further. The dissociation constant of G1.30.B10 to WT FAH was measured by NMR spectroscopy using the saturation transfer difference experiment [19] (STD, Figure 3B). The K_D_ = 235 µM found is modest but consistent with the reduced molecular weight of the fragments. Yet, a cell viability assay (Appendix A) indicates that no toxicity was found at the active concentrations.

### 2.5. The Compound G1.30.B10 Stabilizes the Dimeric Species of FAH

We next investigated to what extent the compound G1.30.B10 alters the FAH monomer-dimer equilibrium. FAH undergoes an equilibrium between an inactive monomer and an active dimer, with a dimer dissociation constant around 0.5–1 µM for WT-FAH [11]. FAH monomers tend to aggregate at concentrations above 1 µM and thermal melts monitored by circular dichroism are sensitive to the dimer’s dissociation, since they generate a second apparent melt around 40 °C that results from the monomer precipitation (Figure 1C, black circles). As shown in Figure 1C, in the presence of G1.30.B10, the apparent thermal melt of WT-FAH is reduced in a proportional way to the chaperone concentration. Moreover, the incubation of V166G-FAH with G1.30.B10 significantly reduced the aggregation tendency of the protein over time (Figure 1D). We hypothesize that this stabilization is produced by securing the dimeric species of FAH.

Diffusion-ordered 2D NMR spectroscopy (DOSY), which employs pulsed field gradient (PFG) to encode spatial information, can be used for measuring the translational diffusion coefficient (D), which is both sensitive to molecular weight and shape [20]. As shown in Figure 3C, the DOSY experiment run for V166G FAH dimer at 35 °C results in a log(D) of −9.3, which, according to a previous diffusion study on FAH [11], indicates that the protein equilibrium populates the monomeric form (60%), while, at the same temperature, WT FAH is 90% dimer (logD = −9.7). In the presence of 10 equivalents of G1.30.B10, the dimer population of V166G-FAH increased to 80% (logD = −9.6).

### 2.6. Structural Design of a Second Generation of FAH Activators

We employed methyl-TROSY NMR spectroscopy [21] on a FAH sample selectively labeled with ^13^CH_3_ in the methionine methyl groups to structurally characterize the interaction between this compound and the protein. Methionines M140 and M180, assigned using site-directed mutagenesis, are close to the binding/dimerization sites (Figure 3D) and are good reporters for binding monitoring since they are well resolved in the methyl-TROSY spectrum (Figure 3E). M140 (but no M180) is significantly perturbed in the presence of G1.30.B10, showing a signal splitting that would indicate two different binding events in independent sites, consistent with the docking analysis (Figure 3D,E): (i) a first interaction with the electrostatic and hydrophobic residues near the entrance of the active site and, (ii) a second interaction that implies the insertion of the compound in the active site cavity. This structural model also provides a structural rationale for the empiric observation that G1.30.B10 promotes dimeric species as the compound stabilizes a loop between Y244 and F249 of the opposite monomer subunit that inserts its residues P246 and L247 into the active site.

Considering the proximity between the cavity formed at the entrance and the active site, we then decided to exploit this property to expand the size (and, thus, the chemical efficiency) of the inhibitors. As shown in Figure 3A, we tested a second generation of molecules (G2) that, based on the molecular scaffold of G1.30.B10, expand the molecule to about 11 Å (length of the full cavity), mostly in a symmetric manner. This strategy included the duplication of the pyridine ring (G2.HT1.1), the duplication of the entire molecule (G2.HT1.2), and two larger non-symmetric molecules (G2.HT1.3 and G2.HT1.4). G2-compounds successfully increased the potency of the inhibitors and, for instance, the compound G2.HT1.1 binds FAH with 10-fold more affinity (K_D_ = 20 µM, Figure 3B). Similar affinity (K_D_ = 21 µM) was obtained for G2.HT1.4. Consistently, all the G2-compounds perturbed the M140 ^13^C-Met chemical shift (Figure 3E) and increased the dimer population of V166-FAH (Figure 3C). In fact, DOSY spectroscopy demonstrates that compounds G2.HT1.3 and G2.HT1.1 induce even higher full dimer populations than that observed for apoWT-FAH.

### 2.7. Inhibitors Restore the Normal Phenotype in a CRISPR/Cas9 Cellular Model of HT1

To investigate the effect of the inhibitors (G2-compounds and G1.30.B10) in the tyrosinemia catabolism, we employed CRISPR/Cas9 technology to transform HEK293 cells into human cellular models of HT1 by replacing the endogenous FAH WT by means of the generation of the carrying G337S (Figure 4A). This mutation drives the accumulation of toxic metabolites such as hydroxyphenylpyruvate and succinylacetone. WT-HEK293 and mutant HEK293 FAH-G337S cells were characterized for their tyrosine catabolism properties by measuring the accumulation (or not) of toxic metabolites in the cytosol and in the extracellular media. In fact, NMR spectroscopy of the extracellular media can distinguish between WT HEK293 cells (reference metabolism) and the HEK293 FAH-G337S cells (that accumulate by-products), as shown in Figure 4B,C. Remarkably, the tested compounds (at 50 µM for G1.30.B10 and 25 µM for G2.HT1.1) can reduce the accumulation of the toxic by-products, largely restituting the phenotype of the WT HEK293 cells.

## 3. Discussion

HT1 is a systemic pathology that can be detected in multiple tissues, such as lymphocytes and fibroblasts. Yet, the liver and kidney are the two main organs affected in patients with HT1 [22], leading to hypophosphatemic rickets and hepatocellular carcinoma due to the mutagenic effects of accumulated toxic metabolites [23]. The use of nitisinone combined with the low tyrosine and phenylalanine intake has proven to be very efficient in preventing HT1 progression, and is the first line of treatment in children [24]. Still, despite the improvement of liver function and portal hypertension with this treatment, nitisinone does not correct the abnormal expression of the gene in the liver, it does not reduce the frequency of hepatocellular carcinoma, and it cannot reverse the already established liver lesions such as oncogenic mutations, hepatic dysplasia, and fibrosis [3]. Altogether, new therapeutic intervention lines are desirable to treat this severe disease.

Our approach to address this challenge is the search for pharmacological chaperones that stabilize the bioactive conformation of the FAH enzyme. Recent results in our laboratory demonstrate that only the dimeric species of FAH shows catalytic activity, in good agreement with a previous structural analysis [8,15]. Yet, such conformation is only marginally stable, since the monomeric species tends to aggregate irreversibly, a mechanism that results in a reduced intracellular concentration of protein [11]. For WT-FAH, at physiological temperature, the dimer population lasts long enough to exert its function. However, most of the deleterious mutants accelerate the aggregation pathway, reducing the protein homeostasis and constituting one of the main mechanisms of pathogenicity associated with HT1. Specifically, 18 out of the 19 studied mutants impair the kinetic stabilization of FAH, and 10 of them directly affect the monomer–dimer equilibrium [11].

Small-molecule pharmacological chaperones function by binding to the target protein and thus stabilizing it. For many misfolding-prone proteins, ideally, pharmacological chaperones should be able to increase protein function by elevating the concentration of folded functional protein. This is the case for the compounds that we found to bind and stabilize FAH: in all cases, productive association implies stabilization of the dimer species and subsequent increase enzymatic activity and, most important, revert the toxic phenotype in a bona fide cellular model of the disease generated by CRISPR/Cas9. Moreover, structural analysis by NMR provides a molecular rationale of the binding of the compounds in a very homogeneous druggable site. The proximity (and partial overlap) between the catalytic and the approximation cavity provides an explanation for the fact that most of the tested compounds also become inhibitors at large concentrations.

Finally, it is important to emphasize that we also addressed the shortcomings of effective models for HT1 by using CRISPR/Cas technology to generate a eukaryotic cell line that introduces a well-known deleterious mutation (FAH-G337S). The cellular model properly mimics the disease since it accumulates succynilacetone and hydroxyphenylpyruvic acid, as determined by NMR spectroscopy. Importantly, cell treatment with the compounds presented here largely abrogates the abnormal phenotype, demonstrating that the compounds act as pharmacological chaperones in cellulo.

## 4. Materials and Methods

### 4.1. Computational Analysis

Ligand docking simulation was accomplished using the open-source programs for AutoDock Vina [25] and AutoDock4.2 in conjunction with AutoDock Tools (ADT) [26,27]. The *Mus musculus* FAH structure (PDB: 1QCN) [15] was used as a protein receptor. Molecular dynamics simulations on the protein receptor were performed as previously described [28]. A ligand library of 20,000 small-compounds was retrieved in SMILES format from the ZINC database [29]. Ligand structure prediction, optimization, and refinement was achieved through minimization by applying CGs with a united force field by employing 25 × 10^6^ steps using Open Babel package (v 2.4.0). A grid centered on the protein of 94 × 93 × 85 Å in the x, y, and z directions was built. The virtual screening results were analysed using in-house MATLAB scripts and FAH druggable sites were visualized using PyMOL. A second run of molecular docking was performed identically using the Maybridge Ro3 Fragment Library containing 2500 selected small compounds (MW < 300) ensuring diversity and pharmacophoric content (Generation 1).

### 4.2. Protein Production and Purification

Protein production was performed using standard protocols previously described [11], with the exception that FAH and glutathione transferase zeta 1 (GSTZ1) were grown at 37 °C for 16 h, while homogentisic acid dioxygenase (HGD) was grown at 25 °C until an OD_600_ of 0.6 to prevent the formation of inclusion bodies. Protein purification was obtained by Ni-NTA resin (Invitrogen^®^) followed by size exclusion chromatography (Hi-Load 26/60 Superdex 75 and 200 for FAH, GE Healthcare) and eluted in 20 mM Tris, 300 mM NaCl, pH 8 for FAH; 20 mM Tris, 500 mM NaCl pH 7.0 for HGD and 5 mM HEPES pH 7.0 with 5% (*v/v*) glycerol for GSTZ1. Protein quantification was performed by spectrophotometrical measurement (E_FAH_(280 nm) = 55,550 M^−1^ cm^−1^; E_HGD_(280 nm) = 69,280 M^−1^ cm^−1^ and E_GSTZ1_(280 nm) = 19,300 M^−1^ cm^−1^).

### 4.3. Chemical Libraries

A commercially available chemical library, the Maybridge Ro3 Fragment Library (Maybridge, Fisher Scientific Int., Portsmouth, NH, USA), was used. Compounds, originally at 200 mM in DMSO, were diluted to 1.2 mM in 50% dimethyl sulfoxide (DMSO) and 50% phosphate buffered saline (PBS) and diluted again in assay media for a final concentration of 60 μM in the screen. The final percentage of DMSO in each assay well, including all control wells, was 0.89%.

### 4.4. Protein Thermal Shift Assay

The protein thermal shift assay (PTS) uses a non-specific dye which, in the presence of a native protein, is naturally quenched, whilst it is released and starts to fluoresce when the protein starts to denature in response to an increase in temperature. PTS assays were performed in a Real Time PCR instrument (ViiA™ 7 Real-Time PCR System; Applied Biosystems, Foster City, CA, USA), in 384-well plates. Plates loaded with 1X Dye (Protein Thermal Shift^TM^ dye kit, Applied Biosystems), 8 µM of D233V-FAH and 60 µM of each compound, were heated in a temperature ramp from 25 to 90 °C, with a heating rate of 1 °C/min. Each experiment was done in quadruplicate in reaction volumes of 15 µL/well, also including controls without compounds. Fluorescence data (excitation wavelength 580 nm and emission wavelength 623 nm) were processed using in-house Matlab scripts. Comparisons of the resulting T_m_ values with respect to the control were used to select potential pharmacological chaperones.

### 4.5. FAA Synthesis and Purification

The synthesis of fumarylacetoacetate was based on Fernández-Cañón and Peñalva’s 1997 published method [30]. First, HGD was activated for 30 min with 20 mM KHPO_4_ pH 7.4, 2 mM ascorbic acid and 1mM FeSO_4_ at room temperature. The solution was then mixed with 20 mM KHPO_4_ pH 7.4, 2 mM ascorbic acid, 0.2 mM FeSO_4_ and 2 mM homogentisic acid. The reaction was incubated at 37 °C for 3 h to achieve full conversion of homogentisic acid into 4-maleylacetoacetic acid. GSTZ1 was then incubated with 20 mM KHPO4 pH 7.4, 2 mM ascorbic acid and 1mM reduced glutathione (Sigma Aldrich) at room temperature. Once the HGD reaction had finished, 10% (*v/v*) of volume reaction of 10% (*w/v*) metaphosphoric acid was poured on ice and left for 10 min to stop the reaction and precipitate the enzymes. Solution was then centrifuged for 10 min at 14,000 rpm, and the supernatant was neutralized with 4M KOH. Activated GSTZ1 was poured directly into the solution in a proportion of 1:10 and the mixture was left for 2 h at 37 °C and stopped by adding 10% (*v/v*) of the total volume of 10% (*w/v*) metaphosphoric acid and was left on ice for ten minutes. FAA purification was performed in a Shimadzu HPLC equipped with SIL20AC HT multisampler, CTO-10AS VP, LC-20AD, CBM-20A and SPD-M20A modules. The fumarylacetoacetate solution was acidified with acetic acid at pH 3 and loaded into a BDS Hypersil C18 column (250 × 3 mm Teknokroma, Thermo Scientific) in an acetic acid-water (A) and acetonitrile (B) gradient. The gradient program was set at 1 mL/min as follows: 0–10 min; from 0% (*v/v*) to 20% (*v/v*) B; 10–10.1 min; from 20% (*v/v*) to 0% (*v/v*) B; and remained for 10 min at 0 % (*v/v*) B to re-equilibrate the column before the next injection. Under these conditions, the FAA peak elutes at 10 min. The fumarylacetoacetate concentration was quantified using the molar extinction coefficient of 13,500 M^−1^ cm^−1^ at 330 nm.

### 4.6. Enzymatic Assay

The enzymatic activity of FAH measures the disappearance of FAA monitored at 330 nm by an HPLC equipped with a diode array. The assay was performed in 50 mM sodium phosphate buffer pH 7.4 in a dry bath incubator at physiological temperature for several minutes (between 2 and 10 min). The optimal time was determined for each batch of purified protein. Samples were then frozen to stop the reaction and measured spectrophotometrically to assess the remaining amount of FAA. Specific activity is then calculated by comparing to the amount of FAA in the control experiment (without compound).

### 4.7. NMR Spectroscopy Experiments

All NMR experiments were performed on a Bruker Avance III 800 MHz NMR Spectrometer equipped with triple resonance ^1^H, ^13^C, ^15^N-cryoprobe. All experiments were performed at between 298 and 308 K. Chemical shifts are given in ppm with respect to 4,4-dimethyl-4-silapentane-1-sulfonic acid (DSS) as an internal reference. The 2D-DOSY experiments were carried out by recording 128 scans for each gradient step, with the in-house tdDOSYccbp.2D pulse sequence, which is based on the standard dstebpgp3s pulse sequence from Bruker [31]. An in-house sequence uses bipolar gradients pairs to refocus chemical shift effects and compensate for chemical exchange that could lead to oscillations in the decay curve [32,33]. The experiments were run using linear gradient of 32 steps between 2% and 95%, a diffusion time (Δ) of 0.2 s, and the length of the square diffusion encoding gradient pulses (δ) of 3 ms. The standard Bruker protocol was used for processing in TopSpin 3.2 software. The fitting of the diffusion dimension in the 2D-DOSY spectra was achieved using an exponential fit. In the processed 2D spectrum, the y-axis shows the values of logD (D = diffusion coefficient (m^2^/s)), while the x-axis shows the ^1^H chemical shift (in ppm scale).

Compound titration to FAH was performed by monitoring the change in the saturation transfer difference (STD). In an STD experiment, the amplification STD factor A_STD_ is defined as:(1)$$A_{STD}=\frac{\propto_{STD}\cdot \left[L\right]}{K_{D}+\left[L\right]}$$
where α*_STD_* is the maximum amplification factor and [*L*] is the ligand concentration. The protein concentration was set to 30 µM while the ligand concentration varied between 0 and 900 µM. All the experiments used 3 s saturation time at a position of −0.5 ppm.

2D ^13^C^−1^H heteronuclear multiple-quantum correlation (HMQC) NMR spectra (methyl TROSY) were recorded using 200 increments and processed without using linear prediction.

### 4.8. Mammalian Cell Culture and Transfection

Human fibroblastoid M1 cell line [18] was grown in complete DMEM medium (Dulbecco’s modified Eagle’s medium) supplemented with 10% (*v/v*) fetal bovine serum (FBS), 0–1 mg/mL streptomycin, and 100 units/mL penicillin at 37 °C in a humidified atmosphere of 5% CO_2_-95% air. To maintain and amplify cells in a proper confluence, they were counted by means of an automated cell counter (CountessTM, Life technologies) that performs cell count and viability calculations (from alive, dead and total cells) using Trypan blue staining. Transfection of a plasmid carrying the FAH-GFP constructs (pCMV6-AC-GFP) in M1 cells were performed using X-tremeGENE^TM^ HP DNA Transfection Reagent (Roche) following the manufacturer’s instructions.

### 4.9. GFP Fluorescence Detection

Cells were seeded up to 50–80% confluence on glass coverslips for 24–48 h. Then, cells were fixed for 10 min with 2% (*v/v*) formaldehyde in PBS and then washed twice with PBS. Finally, the coverslips were mounted onto glass slides on Fluoromont G (Southern Biotechnology Associates, Birmingham, AL) containing 0.7 g/mL DAPI to stain DNA (nucleus) to GFP direct detection. All procedures were carried out at room temperature. All immunostainings were analyzed in a fluorescent microscope (Zeiss Axiovert 200).

### 4.10. Cell Viability Assays (MTT Assay)

Cell lines were inoculated in a 96-well cell culture plate (100 μL/well) with different compounds from commercial libraries at a series of 1/2 dilutions from 1 to 0 mM, with 3 parallel wells in each group. Cells were cultured in an incubator at 37 °C, saturated humidity and 5% CO_2_ for 24 h, and then washed twice PBS1X. Ten microliters of 3-(4,5-dimethylthiazol-2-yl)-2,5-diphenyltetrazolium bromide (MTT, 0.5 mg/mL) were added into each well and the cells were cultured again for 1 h. All media were then removed and 200 μL of DMSO was added to each well and mixed. The absorbance at wavelength 550 nm was measured by a microplate reader. IC_50_ was calculated by linear regression as described by Nevozhay [34].

### 4.11. Generation of FAH-G337S Mutant HEK Cells by CRISPR/Cas9-Mediated Genomic Edition

The FAH missense mutation (c.1009G > A) was introduced by targeted homologous recombination at the endogenous FAH locus in human embryonic kidney (HEK) cells. HEK293 cells were co-transfected using Lipofectamine 3000 (Thermo Fisher Scientific, Waltham MA) with the *p*.X459-FAH -sgRNA plasmid (2.5 µg/well) and single strand oligonucleotide (ssODN) (7.4 µg) containing the FAH-G337S mutation as well as flanking sequences on each side of the targeted exon. The pX459 plasmid was obtained from Addgene (Ref. 62988) and expresses Cas9 from *S. pyogenes*, puromycin resistance gene and cloning backbone for sgRNA. The targeted sequence was designed next to a NGG sequence named protospacer adjacent motif (PAM, N can be any nucleotide). Additionally, a silent substitution was introduced in the PAM sequence of the ssODN in order to prevent secondary cleavage after homology directed repair (HDR) at the FAH locus and enhance recombination efficiency. Transfected cells were cultured in DMEM supplemented with 10% fetal bovine serum (FBS) and 1% antibiotics at 37 °C and 5% CO_2_ for 22 h, after which media were replaced and RS-1 reagent (Sigma, Saint Louis, MO, USA) was added at 10 µM final concentration in order to stimulate HDR. Cells were incubated under the same conditions for another 5 h and then puromycin was added to media in a final concentration of 1 µg mL^−1^, in order to select only the transfected clones. Cells were maintained under these conditions for another 2 days, followed by a limiting dilution sub-cloning assay on 96-well plates. Completely isolated cells were progressively grown and passed to bigger dishes in order to obtain monoclonal cell lines. Genomic DNA was extracted from each of these clones and targeted exons were PCR-amplified for FAH gene sequencing to characterize genome editing at the molecular level. As HEK cells are triploid on average, the sequencing analysis is consistent with two alleles resulting from HDR modification were detected in each HEK293-FAHG337S cellular clone, while the third one remained wild type. Successfully modified clones were selected for their phenotypic characterization. Consistently, FAH-targeted clones exhibited accumulation of succynilacetone and hydroxyphenylpyruvic acid, as determined by NMR spectroscopy.

### 4.12. Metabolite Extraction for NMR

Culture media were mixed with cold methanol (stored at −80 °C) in a 1:2 media:methanol manner and left at −20 °C for 30 min to allow for protein precipitation. After centrifugation at 16,000× *g*, 4 °C for 20 min, the supernatant was dried in a speed vacuum system and stored at −20 °C. For NMR measurement, samples were reconstituted in D_2_O.

### 4.13. Statistical Analyses

The results are expressed as averages and standard deviations.

## Figures and Tables

**Figure 1 ijms-22-01789-f001:**
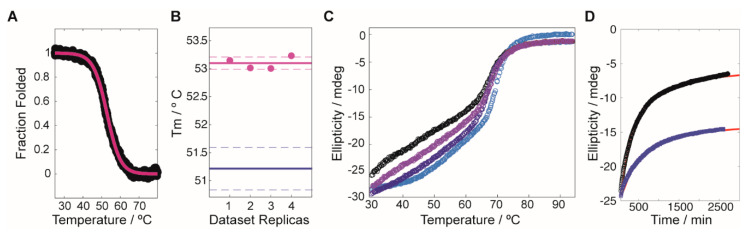
Compound induced stability experiments on fumarylacetoacetate hydrolase (FAH). (**A**) Typical profile for the stability screening, where the black circles correspond to the experimental data (converted to fraction folded) and the red line is the best fit to the dataset. (**B**) Reproducibility of the Tm values obtained by using the linear extrapolation model on the sigmoidal decays. The purple circles correspond to the experimental data, with an average value/uncertainty represented by the solid/dashed lines. Equivalent values for the reference protein (in the absence of compound) are shown in blue. (**C**) Thermal denaturation experiment for WT-FAH in the absence (black circles) and in the presence of increasing concentrations of the compound G1.30.B10 (light purple, 15 eq.; dark purple, 30 eq.; blue, 45 eq.). (**D**) Aggregation tendency of V166G-FAH over time (monitored by the ellipticity at 222 nm), in the absence (black circles) and in the presence (blue squares) of 15 eq. of G1.30.B10. The red lines correspond to the best exponential fittings to the experimental datasets.

**Figure 2 ijms-22-01789-f002:**
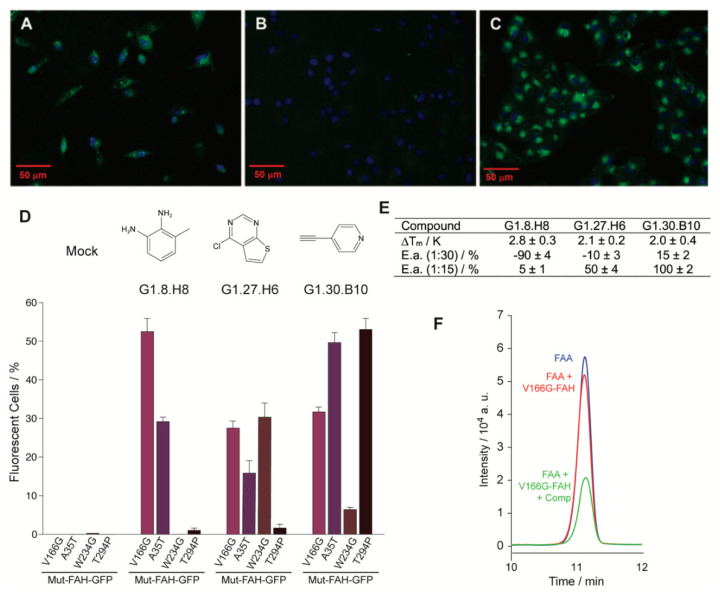
Eukaryotic and in vitro assays for a selected set of compounds. (**A**–**C**) Representative fluorescence microscopy images for the eukaryotic cellular assay. M1 cells incubated with (**A**) the proteasomal inhibitor MG132, (**B**) a compound of the library showing no effect on FAH homeostasis and (**C**) a compound that increases protein accumulation in the cytosol. (**D**) Percentage of positive cells for a set of selected compounds and for the four different mutant constructs under consideration: V166G-FAH-GFP, purple bars; A35T-FAH-GFP, violet bars; W234G-FAH-GFP, brown bars and T294P-FAH-GFP, black bars. (**E**) Change in the thermal stability (T_m_) for the selected set of compounds and percentage of inhibition or activation as compared to the reference reaction for the set of compounds in the enzyme activity assay (E.a.), in the presence of 15 and 30 equivalents of a given compound as indicated. Error bars are obtained from triplicate measurements. (**F**) Region of the HPLC chromatogram corresponding to the fumaryl acetoacetate (FAA) elution. Legend: control experiment without enzyme (blue line), enzymatic reaction (red line), enzymatic reaction in the presence of compound (green line).

**Figure 3 ijms-22-01789-f003:**
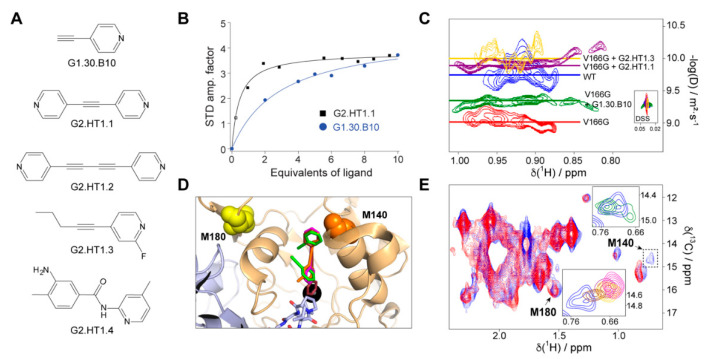
Inhibitors and chaperones found in this study. (**A**) Molecular formulas for the compounds under consideration. (**B**) Representative examples of the binding affinities as determined by saturation transfer difference spectroscopy. Blue circles and black squares correspond to the experimental data for G1.30.B10 and G2.HT1.1, respectively. The solid lines reflect the best fitting of the experimental data to Equation (1). (**C**) Diffusion-ordered 2D NMR spectroscopy (DOSY) experiment performed on WT-FAH (blue), the unstable variant V166G-FAH (red), and the same protein in the presence of 10 eq. of G1.30.B10 (green), G2.HT1.1 (purple) or G2.HT1.3 (yellow). The diffusion coefficient is proportional to the molecular weight of the particle. (**D**) Structural model of the complex between FAH and a selected set of compounds including G1.30.B10 (blue), G2.HT1.2 (purple), and G2.HT1.3 (red). The longer compounds occupy both the active site and the approximation cavity, while G1.30.B10 independently binds to each of these sites. (**E**) Methionine methyl region of the ^1^H,^13^C correlation spectrum and chemical shift perturbation in the presence of the compounds. Both insets correspond to M140, in the absence (light blue) and in the presence of compounds, using the same color code. Notice the signal splitting in the presence of G1.30.B10 (upper inset).

**Figure 4 ijms-22-01789-f004:**
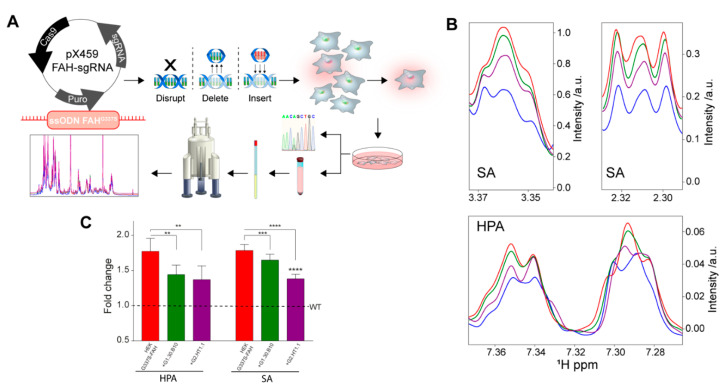
A cellular model of HT1. (**A**) Scheme for the generation of an HT1 cellular model, based on the genomic modification (G337S in the *fah* gene) using the CRISPR/Cas9 technology. (**B**) Regions of the ^1^H-NMR spectrum of the extracellular media in contact with: WT-HEK cells (blue), G337S-FAH-HEK cells (red), or the latter but incubated with G1.30.B10 (green) or G2.HT1.1 (purple). The cellular model adequately reflects the phenotype of HT1 patients since it accumulates succinylacetone (SA) and hydroxyphenylpyruvate (HPA), among other metabolites. (**C**) Bar plot with the changes in SA and HPA, quantified from the 1H-NMR spectrum. *P*-values of < 0.01, 0.001 and 0.0001 are represented by **, *** and **** respectively.

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
