# Peer review of "Therapeutic Targeting of Fumaryl Acetoacetate Hydrolase in Hereditary Tyrosinemia Type I"

_ijms, 2021, doi:10.3390/ijms22041789_

Round 1
Reviewer 1 Report
The authors described the discovery of small molecule modulator against fumaryl acetoacetate hydrolase in hereditary tyrosinemia type I.
The process to identify the desired compounds are well-written from druggable assessment of dimerization domain to identification of several small momolecules.
Even though the evidence of structural information is still paucity, whole story is interesting for readers.
Thus, I would recommend that this manuscript could be publication in International Journal of Molecular Sciences, as it stands.
Author Response
Thank you very much for your comments and appreciation of our work.
Reviewer 2 Report
The manuscript describes the discovery of chemical modulators of fumarylacetoacetate hydrolase (FAH). A deficiency in human FAH leads to hereditary tyrosinemia type I (HT1) and the study aim is to develop a new drug for this genetic disease. Chemical modulators stabilize
the active dimeric form of this enzyme. In the first step, the authors used molecular docking calculations to prove that the dimerization interface is druggable. Next, they used a protein thermal shift assay to identify molecules from a commercially available chemical library of 2500 molecules that increase the thermostability of the D233V-FAH mutant. The same library was used for functional assay with V166G-FAH-GFP protein. Finally, the set of compounds that stabilized FAH was tested for modulation of FAH catalytic activity in vitro using V166G-FAH mutant.
The selected compound G1.30.B10 enhanced the activity of V166G-FAH at all ratios, indicating less competition with the substrate for binding. Diffusion-ordered 2D NMR spectroscopy (DOSY) showed that this compound increases the dimer population of V166G-FAH from 40 to 80%
The authors employed methyl-TROSY NMR spectroscopy to structurally characterize the interaction between this compound and the protein. Together with the docking analysis, this demonstrated that compound G1.30.B10 promotes dimeric species of the FAH stabilizing a loop between Y244 and F249 of the opposite monomer subunit. Using this data the authors designed a set of new molecules expanding their size to the length of the full cavity. DOSY spectroscopy demonstrated that two of newly designed compounds induce even higher full dimer populations of V166G-FAH than the one observed for apoWT-FAH. In an additional experiment the authors demonstrated that the newly designed inhibitors restore the normal phenotype in a CRISPR/Cas9 cellular model of HT1.
The paper is well written and the research design is appropriate. I have only a single question: the authors performed molecular docking using the same 2500 compounds library which was used for experiments, but did not discuss the results. How well the results of docking correlates with the experimental results ?
Author Response
Thank you very much for your comments and for your consideration towards our work. Regarding the docking efficiency, 90% of the molecules that gave positive outcome in the stability and functional tests were correctly docked at the binding site. We have added the following sentence in the text (line 145):
Interestingly, 90% of the compounds that gave positive results in both assays were also docked at the active site of the enzyme.